# Derepression may masquerade as activation in ligand-gated ion channels

Christian J. G. Tessier [1], Johnathon R. Emlaw [1], Raymond M. Sturgeon [1] & Corrie J. B. daCosta [1]✉

Agonists are ligands that bind to receptors and activate them. In the case of ligand-gated ion channels, such as the muscle-type nicotinic acetylcholine receptor, mechanisms of agonist activation have been studied for decades. Taking advantage of a reconstructed ancestral muscle-type β-subunit that forms spontaneously activating homopentamers, here we show that incorporation of human muscle-type α-subunits appears to repress spontaneous activity, and furthermore that the presence of agonist relieves this apparent α-subunit-dependent repression. Our results demonstrate that rather than provoking channel activation/opening, agonists may instead 'inhibit the inhibition' of intrinsic spontaneous activity. Thus, agonist activation may be the apparent manifestation of agonist-induced derepression. These results provide insight into intermediate states that precede channel opening and have implications for the interpretation of agonism in ligand-gated ion channels.

An enduring goal is to understand how agonists activate their receptors[1]. For ligand-gated, or 'agonist-activated', ion channels, such as the nicotinic acetylcholine receptor (AChR), agonist binding culminates in activation/opening of an intrinsic ion-conducting pore[2]. Early models postulated that agonists are effective at opening the channel because they have a higher affinity for the open state than the closed state, and thus their binding shifts the equilibrium towards the open state[3]. In a landmark study, del Castillo and Katz proposed that agonism could be explained by a two-step process, where the agonist first binds to an inactive receptor, that then isomerises to an active/open conformation[4]. Within this framework, the ability of a given agonist to catalyse the isomerization step, and open the channel, was encoded in the chemical structure of the agonist, as well as its interactions with the receptor-binding site. Full vs. partial agonists differed in their ability to catalyse the isomerization step, with full agonists being more effective than partial agonists[5].

The advent of the patch clamp technique and the development of single-channel analysis methods have made detailed investigations of the activation mechanism possible[5–7]. Early single-channel experiments were consistent with del Castillo and Katz's two-step mechanism, however, to account for the two agonist sites on the AChR, the original scheme had to be extended to include two agonist-binding steps instead of one[8]. The emerging single-channel data also revealed additional complexity, including a class of brief closings, originally called 'nachschlag shuttings', which interrupted bursts of agonist-activated openings[9]. In some cases, the duration of these closings appeared to be agonist-independent, necessitating the introduction of an additional closed state in a more intricate scheme[10]. These observations, combined with studies of the related glycine receptor[11] and continued technical improvements[12], ultimately led to more elaborate mechanisms that incorporated intermediate closed states, termed 'flipped' or 'primed', which preceded channel opening[3,13]. These findings placed the origins of agonism before channel opening in the overall activation process[3], and demonstrated that the ultimate opening and closing rates of the AChR were independent of the agonist used to elicit them; a somewhat counterintuitive finding for a channel 'activated' by agonist.

Despite the explanatory power of intermediate closed states, they have been inferred solely from kinetic analysis of single-channel measurements, and thus they remain largely phenomenological[14]. What is 'flipping'? What is 'priming'? To gain further insight we have taken advantage of a reconstructed ancestral muscle-type AChR β-subunit, called 'β_{Anc}', which at the amino acid level shares more than 70% sequence identity with the human β-subunit[15]. In

[1]Department of Chemistry and Biomolecular Sciences, Centre for Chemical and Synthetic Biology, University of Ottawa, Ottawa, ON, Canada.
✉e-mail: cdacosta@uottawa.ca

addition to forming hybrid ancestral/human AChRs, where $\beta_{Anc}$ replaces both the human $\beta$- and $\delta$-subunits[16], $\beta_{Anc}$ also forms homopentamers[17]. These $\beta_{Anc}$ homopentamers 'prime' before opening spontaneously and, reminiscent of the wild-type AChR, display steady-state burst behaviour with *nachschlag*-like shuttings (Fig. 1a)[17]. For $\beta_{Anc}$ to occupy both the $\beta$- and $\delta$-subunit positions in hybrid AChRs, the (+) and (−) interfaces of $\beta_{Anc}$ must not only be compatible with each other, but also with the corresponding interfaces of their bracketing $\alpha$-subunits (Fig. 1a). This suggests that heteropentamers composed of $\beta_{Anc}$ and the human $\alpha$-subunit should be possible (Fig. 1b). If so, this would provide an opportunity to probe the function of the $\alpha$-subunit, and specifically its role in agonism, in a unique context.

Here, we show that $\alpha/\beta_{Anc}$ heteromers are indeed viable, and that incorporation of the human $\alpha$-subunit leads to an apparent $\alpha$-subunit-dependent repression of spontaneous activity, which can be relieved by the addition of agonist. These findings provide insight into the origin of intermediate states that precede ligand-gated ion channel opening, and demonstrate how 'agonist activation' may instead be the apparent manifestation of agonist-induced derepression.

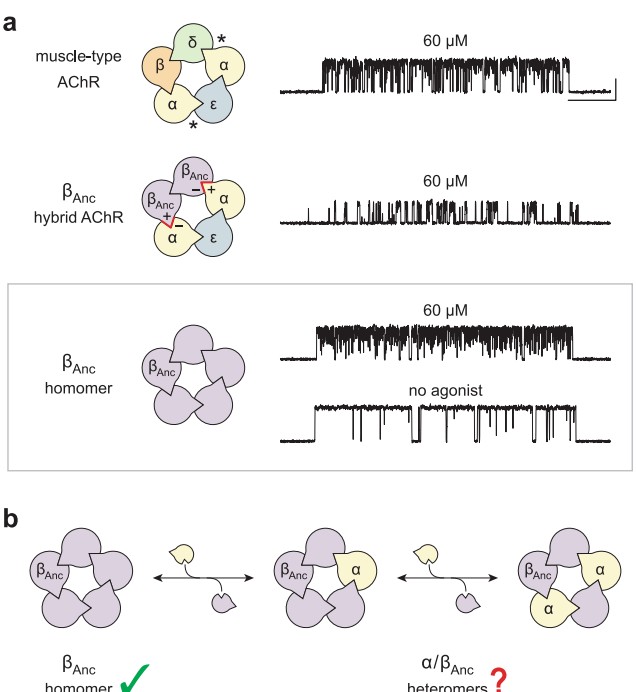

**Fig. 1 | Subunit composition and single-channel activity of wild-type and $\beta_{Anc}$-containing acetylcholine receptors. a** Subunit stoichiometry and arrangement of the human adult muscle-type acetylcholine receptor (muscle-type AChR; *top*), where the agonist-binding sites at the $\alpha$-$\delta$ and $\alpha$-$\varepsilon$ subunit interfaces are indicated with asterisks (*). A reconstructed ancestral $\beta$-subunit ($\beta_{Anc}$; purple) forms hybrid acetylcholine receptors ($\beta_{Anc}$ hybrid AChR; *middle*) where $\beta_{Anc}$ substitutes for the human $\beta$-subunit ($\beta$; orange) and supplants the human $\delta$-subunit ($\delta$; green). For this to happen, the principal (+) and complementary (−) interfaces of $\beta_{Anc}$ must be compatible with each other, as well as with the corresponding interfaces of their bracketing $\alpha$-subunits (red highlights). $\beta_{Anc}$ also forms homomers (*bottom*; boxed), which open spontaneously (no agonist), and whose single-channel activity in the presence of agonist mirrors that of the muscle-type acetylcholine receptor. Recordings were obtained in the cell-attached patch configuration with a constant applied potential of −120 mV and filtered with a 5 kHz digital Gaussian filter. Unless otherwise indicated, all recordings were acquired in the presence of 60 µM acetylcholine, where openings represent inward cation currents, and are shown as upward deflections. The scale bar (20 ms, 10 pA) aligned to the top trace applies to all traces. **b** The compatibility of $\beta_{Anc}$ with bracketing $\alpha$-subunits in hybrid AChRs predicts that $\alpha/\beta_{Anc}$ heteromers are viable.

## Results

### The human $\alpha$-subunit coassembles with $\beta_{Anc}$

We previously showed that $\beta_{Anc}$ forms spontaneously opening homopentamers that at the single-channel level share hallmarks of human muscle-type AChR activity, including steady-state burst behaviour with *nachschlag*-like shuttings[17]. At concentrations of acetylcholine approaching saturation of the adult human AChR, bursts of single-channel activity from the two types of channels are essentially indistinguishable (Fig. 1a). This reflects both the high open probability of bursts of $\beta_{Anc}$ homopentamer activity that is independent of agonist, as well as the similar extent of open-channel block by acetylcholine in the two types of channels[17].

To determine if the human muscle-type $\alpha$-subunit can coassemble with $\beta_{Anc}$ to form $\alpha/\beta_{Anc}$ heteromers, we measured cell-surface binding of radiolabeled $\alpha$-bungarotoxin ($\alpha$-Btx), an AChR competitive antagonist that can bind exclusively to determinants on the $\alpha$-subunit[18]. When cells were cotransfected with cDNAs encoding the four human muscle-type AChR subunits, robust cell-surface binding of $\alpha$-Btx was detected (Fig. 2a). By contrast, cells transfected with cDNA encoding only the human $\alpha$-subunit displayed little or no binding of $\alpha$-Btx, indicating that on its own, the $\alpha$-subunit is incapable of forming $\alpha$-Btx-binding sites that express on the cell surface. Similarly, when cells were transfected with cDNA encoding only $\beta_{Anc}$, essentially no binding of $\alpha$-Btx was detected. Given that single-channel activity of $\beta_{Anc}$ homopentamers is readily observed (Fig. 2c, top trace)[17], the simplest interpretation is that despite their robust cell-surface expression, $\beta_{Anc}$ homopentamers do not bind $\alpha$-Btx. Evidently, as with its extant $\beta$-subunit counterparts, $\beta_{Anc}$ lacks essential determinants of $\alpha$-Btx binding. When cDNAs encoding $\beta_{Anc}$ and the human $\alpha$-subunit are transfected together, robust cell-surface binding of $\alpha$-Btx is restored, demonstrating that $\beta_{Anc}$ can shepherd the $\alpha$-subunits and their associated $\alpha$-Btx-binding sites to the cell surface, presumably by incorporating them into channels containing both types of subunits (Fig. 2a).

### Incorporation of $\alpha$-subunits reduces spontaneous activity

To assess the functional consequences of replacing one or more $\beta_{Anc}$ subunit(s) in $\beta_{Anc}$ homopentamers with the human $\alpha$-subunit, we cotransfected cells with cDNAs encoding both types of subunits and then examined the single-channel activity from cell-attached patches in the absence of agonist (Fig. 2b, c). Addition of $\alpha$-subunit cDNA to a transfection mixture containing the same amount of $\beta_{Anc}$ cDNA resulted in an overall reduction in the spontaneous activity of patches. The extent of this reduction related to the amount of $\alpha$-subunit cDNA added, and was best quantified as an $\alpha$-subunit-dependent decrease in the frequency of spontaneous bursts (Fig. 2b).

Co-transfection with $\alpha$-subunit cDNA also led to changes in the burst behaviour of the resulting channels (Fig. 2c, d). In the absence of the $\alpha$-subunit, bursts were uniformly long, with a mean duration of ~10 ms (Fig. 2c, d; top). When cells were cotransfected with $\alpha$:$\beta_{Anc}$ cDNAs at a ratio of 1:1 (by weight), the introduction of the $\alpha$-subunit led to a second class of briefer events, bracketed by relatively long shut periods, which appeared as isolated openings (Fig. 2c; middle, inset *ii*) and contrasted with the bursts of closely spaced openings observed with $\beta_{Anc}$ homopentamers (Fig. 2c; top & middle, inset *i*). The two types of events were clearly distinguished in burst duration histograms (Fig. 2d; middle), where individual openings separated by closings briefer than a specified critical closed duration ($\tau_{crit}$; 2 ms throughout) have been combined into bursts[19]. The heterogeneous burst behaviour of channels in these patches suggests that two types of channels, $\beta_{Anc}$ homopentamers and $\alpha/\beta_{Anc}$ heteropentamers, are present. Consistent with this hypothesis, increasing the ratio of cotransfected $\alpha$:$\beta_{Anc}$ to 10:1 leads almost exclusively to isolated brief openings, essentially eliminating long bursts, and presumably $\beta_{Anc}$ homopentamers. Evidently, cotransfecting the human $\alpha$-subunit with $\beta_{Anc}$ leads to a reduction in

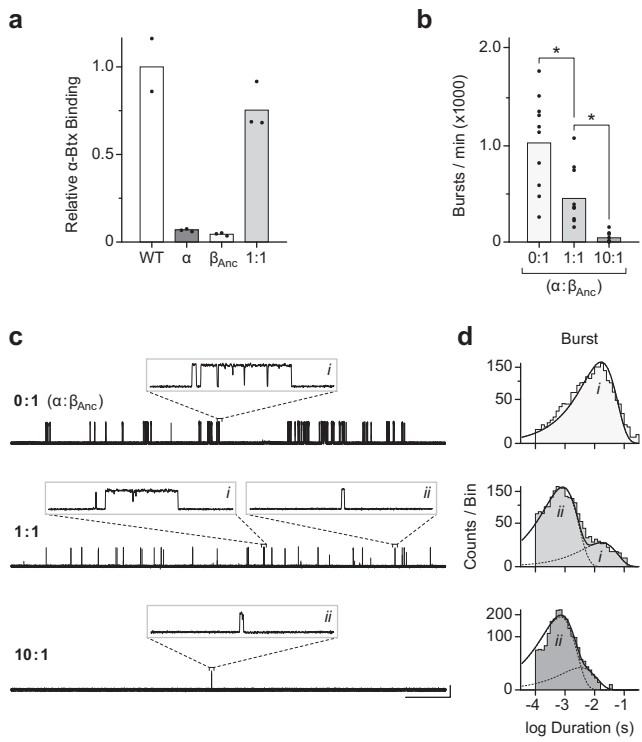

**Fig. 2 | The human α-subunit and β$_{Anc}$ can coassemble to form viable heteromeric channels that open spontaneously. a** Relative cell-surface binding of [$^{125}$I]-α-bungarotoxin (α-Btx) to cells transfected with the full complement of cDNAs encoding the human adult muscle-type acetylcholine receptor subunits (WT; white), the α-subunit alone (α; darkest grey), β$_{Anc}$ alone (β$_{Anc}$; light grey), or a 1:1 (by weight) mixture of the α-subunit and β$_{Anc}$ (1:1; intermediate grey). Bar graphs represent the mean of two or three replicates (shown) from independent transfections, normalised to WT. **b** Frequency of spontaneous bursts of openings from cells transfected with cDNA encoding β$_{Anc}$ alone (0:1; light grey), or with increasing amounts of human α-subunit cDNA (1:1, 10:1; by weight; intermediate and dark grey, respectively), while the amount of β$_{Anc}$ cDNA is kept constant. Bar graphs represent the mean burst frequency averaged across ten separate single-channel patches (individual data points shown in each case), from at least three independent transfections. Difference between mean burst frequency is statistically significant (asterisks), as determined by one-way ANOVA (Tukey's multiple comparison test; α level of 0.05; 0:1 vs. 1:1, $p = 0.0015$; 1:1 vs. 10:1, $p = 0.0247$). **c** Single-channel burst behaviour of patches from cells transfected with different ratios (0:1, 1:1, 10:1; by weight) of α-subunit and β$_{Anc}$ cDNA. In the absence of agonist, recordings were obtained in the cell-attached patch configuration with an applied potential of –120 mV and filtered with a 5 kHz digital Gaussian filter. In each case, openings are upward deflections, with the scale bar (2 s, 10 pA) beside the bottom trace applying to all zoomed out traces, and the inset boxes themselves representing 40 ms and 25 pA. **d** Burst duration histograms (see Methods) were manually fit (solid line) with a minimum sum of exponential components (dashed lines; labelled *i* and *ii*) containing the types of openings shown inset in *c*. Note that the dashed lines in the top panel are hidden by the solid line, as there is only a single exponential component that overlaps perfectly with the overall fit.

both the frequency and duration of spontaneous openings, and thus an apparent repression of the spontaneous activity.

### Reduction in spontaneous activity depends on the number of α-subunits

The above experiments show that the spontaneous single-channel activity of patches containing both β$_{Anc}$ and the human α-subunit is reduced relative to those containing only β$_{Anc}$ homopentamers, and thus that the presence of the α-subunits appears to repress spontaneous activity. Given that the α-subunit only traffics to the cell surface in the presence of β$_{Anc}$, a tacit assumption is that the reduced

spontaneous activity is a result of the α-subunit replacing one or more β$_{Anc}$ subunits in α/β$_{Anc}$ heteromers. To determine how many α-subunits are present in spontaneously opening α/β$_{Anc}$ heteromers we used an electrical fingerprinting strategy, where the α-subunit was tagged with mutations that reduce its contribution to overall single-channel amplitude (Supplementary Figs. 1 and 2). A similar strategy has been employed with tetrameric potassium channels[20], and pentameric ligand-gated ion channels[21–25], including the muscle-type AChR[16]. Incorporation of one or more mutant low conductance α-subunits (α$_{LC}$) would be expected to lead to a progressive decrease in the single-channel amplitude of resulting α$_{LC}$/β$_{Anc}$ heteromers, thereby allowing us to directly register the number of incorporated α$_{LC}$-subunits in individual channels.

Using the wild-type AChR background, we confirmed that channels incorporating α$_{LC}$ had a lower single-channel amplitude, but maintained a similar kinetic profile (Supplementary Fig. 2). This control demonstrated that the conductance altering mutations did not measurably affect other properties of the α-subunit. We then transfected cells with different ratios of α$_{LC}$ and β$_{Anc}$ cDNAs, and measured the distribution of single-channel amplitudes in patches from cells expressing the two types of subunits (Fig. 3). When cells were transfected with cDNAs encoding α$_{LC}$ and β$_{Anc}$, a variety of single-channel amplitudes were observed in each patch (Fig. 3a, b). As expected, the relative proportion of high and low amplitude classes was dependent upon the ratio of transfected α$_{LC}$ to β$_{Anc}$ cDNAs (Supplementary Fig. 3). In mixtures of α$_{LC}$ and β$_{Anc}$, a maximum of only three amplitude classes were detected (Fig. 3c), where the mean amplitude of the highest amplitude class matched that of β$_{Anc}$ homopentamers (Fig. 3c; inset 2, top). Thus, the simplest interpretation is that the intermediate and lowest amplitude classes originated from channels that have incorporated either one or two α$_{LC}$ subunits, respectively. If channels containing more than two α$_{LC}$-subunits were present in these patches, they did not open spontaneously, and thus were not evident in these fingerprinting experiments.

To assess the impact of each α-subunit on the spontaneous activity of β$_{Anc}$-containing channels we first plotted an amplitude vs. duration scatter plot, where each data point represented the amplitude and duration of individual bursts in the fingerprinting experiments (Fig. 3c, inset 1). This analysis showed an apparent correlation between the amplitude and duration of bursts, with lower amplitude bursts being, on average, briefer in duration. This suggested that the incorporation of each successive α$_{LC}$ subunit led to a progressive decrease in burst duration. To quantify this effect, we collected openings whose amplitude was within 1.5 standard deviations of the mean amplitude of each amplitude class (Fig. 3c, inset 2), and then performed dwell time analysis on each set corresponding to spontaneous activity from channels with zero, one, or two α-subunits (Fig. 3d). Consistent with previous results[17], spontaneous openings of β$_{Anc}$ homopentamers were best fit by a single exponential component, and occurred in quick succession, ultimately coalescing into bursts that had a mean duration of ~10 ms. In contrast, channels containing one or two α-subunits exhibited two classes of spontaneous openings, both of which were briefer than the single class of openings in β$_{Anc}$ homopentamers. When one α-subunit was present the longer of these two components was predominant, whereas when a second α-subunit was present the briefest component was predominant. Thus, with incorporation of each successive α-subunit the average duration of openings became briefer and briefer. In addition, when either one or two α-subunits were present, openings appeared as isolated events. This altered burst behaviour was quantified by binning bursts based upon the number of openings they contained, and showed a steep α-subunit-dependent drop off in the number of openings occurring within each burst (Fig. 3d; far right).

### Agonist relieves apparent repression of spontaneous activity

From the perspective of β$_{Anc}$ homopentamers, the above data demonstrate that replacing one or more β$_{Anc}$-subunit(s) with the

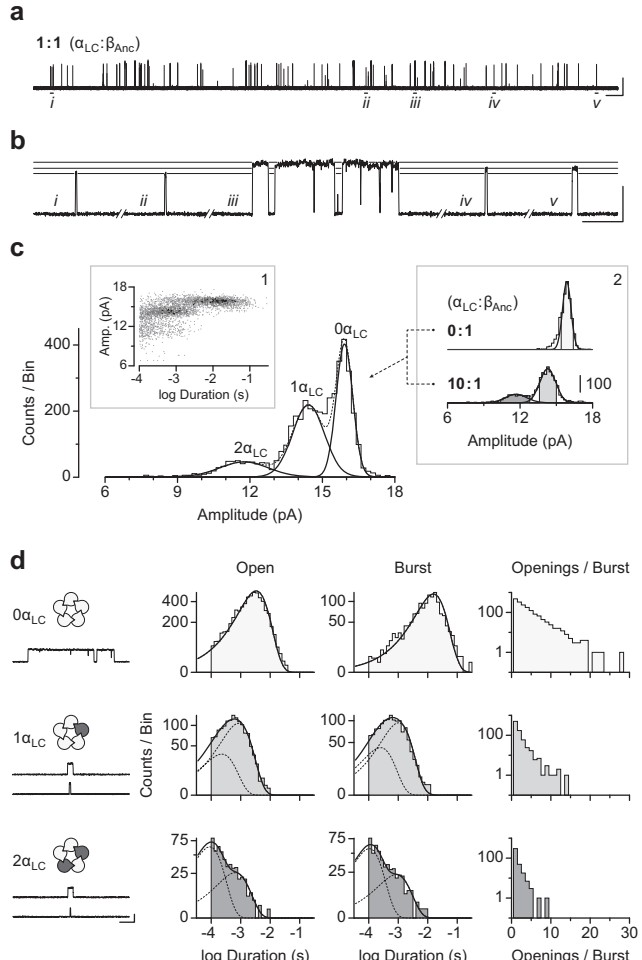

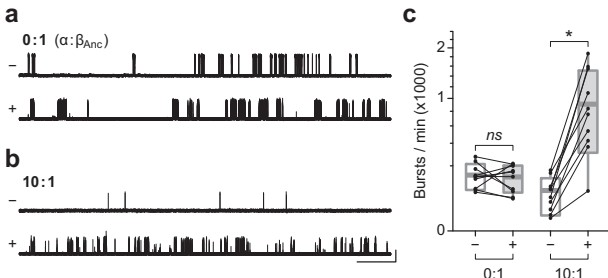

**Fig. 3 | Electrical fingerprinting reveals the contribution of α-subunits to the apparent repression of spontaneous single-channel activity. a** Co-transfection of a cDNA encoding the human α-subunit harbouring reporter mutations that reduce single-channel conductance ($\alpha_{LC}$) with cDNA encoding $\beta_{Anc}$ leads to reduced spontaneous activity, where **b** the amplitude and duration of single-channel events is variable. Indicated openings in (*a; i–v*) are shown expanded in *b*, with sight-lines overlaid to indicate the amplitudes of individual events. Openings are upward deflections, where the scale bars represent 1 s and 10 pA in *a*, and 10 ms and 10 pA in *b*. **c** Combining single-channel bursts from multiple recordings (see also Fig. S3) where cells were transfected at different cDNA ratios (by weight) shows how single-channel amplitudes segregate into three well-defined amplitude classes (overlaid gaussians; solid lines), where each amplitude class corresponds to openings from channels incorporating either 0, 1, or 2 $\alpha_{LC}$-subunits. Plotting the amplitude of individual bursts as a function of their duration reveals an apparent correlation *(inset 1)*. **d** Dwell time analysis of selected bursts whose amplitude is within 1.5 standard deviations of the mean of each amplitude class (shaded regions from *inset 2* in *c*) reveals the contribution of each successive $\alpha_{LC}$-subunit (dark grey subunits in schematics) to open (*left*) and burst (*middle*) durations, as well as in determining the number of successive openings occurring in each burst (*right*). The traces under the schematics (*left*) show openings (upward deflections) representative of each exponential component from visual fits of their corresponding duration histograms (*right*). Scale bar beside the bottom trace in *d* represents 5 ms and 10 pA. Recordings were obtained in the cell-attached patch configuration with an applied potential of −120 mV and filtered with a 5 kHz digital Gaussian filter.

**Fig. 4 | Agonist relieves apparent repression of α/$\beta_{Anc}$ heteromers.** Single-channel activity from cells expressing **a** $\beta_{Anc}$ homopentamers (0:1; α:$\beta_{Anc}$ cDNA; by weight) and **b** α/$\beta_{Anc}$ heteromers (10:1) in the absence (−) and presence (+) of 300 μM acetylcholine. Recordings were obtained in the cell-attached patch configuration with an applied potential of −120 mV and Gaussian filter of 5 kHz. Single-channel openings represent inward cation currents, and are shown as upward deflections. The scale bar beside the bottom trace in *b* applies to all traces and represents 1 s and 10 pA. **c** Comparison of burst frequency in paired recordings from cells expressing either $\beta_{Anc}$ homopentamers (0:1), or α/$\beta_{Anc}$ heteromers (10:1). In each case, paired cell-attached recordings from the same cell were acquired first in the absence (−), and then in the presence (+), of 300 μM acetylcholine in the patch pipette. Box plots represent one standard deviation from the mean, with the internal horizontal line denoting the mean of the 10 recordings in each case. Maximum and minimum values are presented as box plot whiskers. As determined by a two-way ANOVA, the difference between mean burst frequency −/+ 300 μM acetylcholine is statistically significant (α level of 0.05) for the α/$\beta_{Anc}$ heteromers (10:1, $p = 0.0011$; asterisk), but not the $\beta_{Anc}$ homopentamers (0:1, $p = 0.7189$; *ns*).

spontaneous single-channel activity in the absence of agonist. To assess the impact of agonist, we identified cells that had a low, but quantifiable, spontaneous activity in the absence of agonist, and then returned to the same cell to record additional data from a new cell-attached patch, but this time in the presence of 300 μM acetylcholine. We reasoned that this high concentration of acetylcholine might be required to elicit a response given that α/$\beta_{Anc}$ heteromers, with $\beta_{Anc}$ providing the complementary (−) face of the agonist-binding site, lack important residues involved in agonist recognition[16]. Furthermore, by returning to the same cell to acquire paired recordings, first in the absence, and then in the presence of acetylcholine, we were able to limit patch-to-patch variability by controlling for the individual expression level of each transfected cell. The same strategy has been used to quantify calcium potentiation of the human α7 acetylcholine receptor[26]. While extensive open-channel block resulting from the high agonist concentration precludes a meaningful comparison of dwell times (Supplementary Fig. 4), we were able to assess the effect of agonist on burst frequency (Fig. 4). For $\beta_{Anc}$ homopentamers (0:1; α:$\beta_{Anc}$), although there was some variability in the frequency of bursts plus or minus agonist in paired patches from the same cell, the overall frequency of bursts was not significantly different in the presence or absence of agonist. By contrast, in cells transfected with a 10:1 ratio of α:$\beta_{Anc}$ cDNAs (see also 1:1 in Supplementary Fig. 5), paired recordings from the same cell revealed that, in each case, the presence of agonist led to a marked increase in the frequency of bursts. On average, the increase in burst frequency was more than 10-fold, revealing that agonist was able to relieve this aspect of the apparent repression imparted by the α-subunits in α/$\beta_{Anc}$ heteromers (Fig. 4c; Supplementary Fig. 5).

## Discussion

Previously we showed that $\beta_{Anc}$ readily forms homopentameric channels that open spontaneously[17]. Here, we have shown that the human muscle-type α-subunit can coassemble with $\beta_{Anc}$ to form α/$\beta_{Anc}$ heteromers (Fig. 2a). Co-transfection of the α-subunit with $\beta_{Anc}$ led to a reduction in the frequency of spontaneous openings in cell-attached patches (Fig. 2b), as well as alterations in the single-channel behaviour

human α-subunit leads to a reduction in spontaneous single-channel activity. Given that in wild-type AChRs, residues from the α-subunits form the principal (+) face of the agonist-binding sites, we wondered whether this apparent α-subunit-dependent repression could be relieved by the addition of agonist. We began by once again transfecting cells with various ratios of α:$\beta_{Anc}$ cDNAs, and then recorded

consistent with inhibition of the spontaneous activity (Fig. 2c). In both cases, the degree of inhibition was proportional to the amount of α-subunit cotransfected (Fig. 2). Tagging the α-subunits with conductance/reporter mutations allowed us to determine that α/β$_{Anc}$ heteromers containing one or two α-subunits are viable, and open spontaneously (Fig. 3). Dwell time comparison of channels containing zero, one, or two α-subunits revealed that incorporation of each successive α-subunit progressively decreased the duration of both individual spontaneous openings, as well as bursts of closely spaced spontaneous openings (Fig. 3). Burst duration was further decreased through an α-subunit-dependent reduction in the mean number of openings occurring within each burst (Fig. 3). Finally, when both β$_{Anc}$ and the α-subunit were cotransfected, the presence of agonist in cell-attached patches led to a greater than 10-fold increase in the single-channel activity (Fig. 4).

What do these experiments involving unnatural, heterologously expressed, spontaneously opening channels based upon a reconstructed ancestral muscle-type acetylcholine receptor β-subunit tell us about pentameric ligand-gated ion channels found in nature today? In particular, our experiments provide possible insight into the function of the α-subunits in muscle-type AChRs. Relative to β$_{Anc}$ homopentamers, the reduced spontaneous activity of α/β$_{Anc}$ heteromers depends steeply upon the number of incorporated α-subunits, and thus it appears as though the α-subunits have the capacity to repress intrinsic spontaneous activity. However, a similar outcome would be expected if each β$_{Anc}$ subunit imparted an activating presence that is lost in α/β$_{Anc}$ heteromers. Determining the relative contribution of these two possibilities is not trivial, yet the implications of a possible repressive role for the α-subunits are profound. Usually viewed as the principal subunits contributing to agonist binding and activation[27], we've shown that the α-subunits may also have the capacity to repress intrinsic spontaneous activity. While this was demonstrated in an unnatural channel, the incorporated α-subunits were wild-type human α-subunits, raising the possibility that the α-subunits perform the same function in their natural setting, and repress intrinsic spontaneous activity in the heteromeric muscle-type AChR.

From the classic pharmacological perspective, 'activation' refers to the transition of a receptor from an inactive to an active state, following application of agonist[28]. Implied within this definition is that the agonist directly drives the transition of the receptor from an inactive to an active state. For ion channels, their open/ion-conducting conformation is their active state, and thus ion channel activation is the opening of the ion channel driven by an external stimulus. For ligand-gated (or 'agonist-activated') ion channels, such as the AChR, agonists are expected to directly drive the transition from an inactive/closed state to an active/open state[2]. This is precisely why the 'flip' and 'prime' models of agonist activation are surprising. In both models, the ultimate opening and closing rates of the channel are independent of the agonist, begging the question: are agonists bona fide agonists if they do not directly activate the channel?

A repressive role for the α-subunits offers a way to reconcile these counterintuitive findings, and provides insight into phenomenological 'flipped' and 'primed' states. Rather than activating the channel, agonists could instead derepress it. In this framework, the α-subunits repress spontaneous/basal activity, and agonists remove this repressive influence leading to an 'apparent activation', which is in fact the restoration of intrinsic basal activity. To derepress the channel, agonists drive the transition of the channel from a repressed/closed state where openings are unlikely (e.g., 'unflipped' or 'unprimed'), to a derepressed/closed state where openings are more likely (e.g., 'flipped' or 'primed'). An important distinction between derepression and activation, is that in the derepression paradigm, agonists do not drive the transition from an inactive/closed state to an active/open state, but instead drive a transition between two distinct closed states that are both inactive. From this perspective, full vs. partial agonists differ in

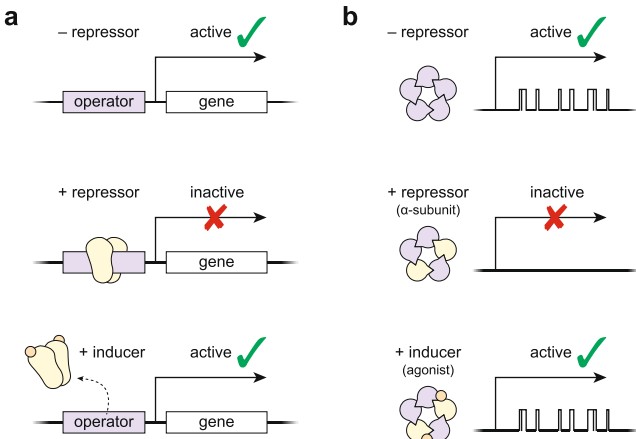

**Fig. 5 | Parallels between induction of gene expression and agonism in a pentameric ligand-gated ion channel. a** Constitutive expression of a gene *(top)* can be repressed by the binding of a repressor (yellow protein) to an operator sequence *(middle)*, which can then be derepressed by the binding of an inducer (orange circle) to the repressor *(bottom)*. **b** Constitutive activity of a pentameric ion channel *(top; purple* β$_{Anc}$*)* can be repressed by incorporation of a repressor protein *(middle; yellow* α*-subunits)*, which can then be derepressed by the binding of an inducer *(bottom; orange agonist)*.

their ability to derepress the channel, but once derepressed, the channel opens and closes in the same way irrespective of the efficacy of the bound agonist. These predictions that (1) agonists drive a transition between two inactive/closed states, and (2) channel opening is independent of agonist efficacy, are both realised in the 'flip' and 'prime' models[3,13].

Mirroring muscle-type AChRs that contain two α-subunits, our fingerprinting experiments detected a maximum of two α-subunits in spontaneously opening α/β$_{Anc}$ heteromers. While we cannot exclude the possibility that three (or more) α-subunits can exist in α/β$_{Anc}$ heteromers, for this to happen, at least two α-subunits would have to occupy neighbouring positions in the resulting heteropentamer. This would require that the (+) and (−) interfaces of the α-subunit be compatible with each other, thereby allowing multiple α-subunits to self-associate. There is no evidence that the human muscle-type α-subunit can self-associate, and consistent with this, we did not detect cell-surface binding of radiolabeled α-Btx when the α-subunit was transfected alone. The simplest interpretation is that α/β$_{Anc}$ heteromers contain at most two α-subunits, where again mirroring muscle-type AChRs, the two α-subunits are separated by at least one intervening subunit (in this case a β$_{Anc}$ subunit). Notably, the presence of two α-subunits in muscle-type AChRs is enough to ensure that spontaneous openings are infrequent[29]. Similarly, two α-subunits in α/β$_{Anc}$ heteromers is enough to reduce spontaneous activity, such that spontaneous openings occur infrequently in comparison to β$_{Anc}$ homopentamers.

The possibility that the α-subunits hold the channel shut, as opposed to being responsible for opening it, also has potential implications for how agonism evolved in this family of proteins. Given that several aspects of wild-type AChR activation are independent of agonist and preserved in β$_{Anc}$ homopentamers, it seems plausible that spontaneous channel activity existed before its regulation by agonist. In such a scenario, spontaneously opening channels evolved a subsequent layer of regulation through accumulating mutations that repressed their constitutive activity, which could then be restored by the presence of bound agonist. Essentially, constitutive/basal activity is inhibited by a repressor protein, in this case the α-subunits, which is then derepressed by the binding of an inducer, in this case an agonist (Fig. 5b). This is consistent with the observation that most AChR mutations alter the intrinsic tendency of the protein to open without

changing the energetic contribution from bound agonist[30]. If regulation by agonist evolved to control pre-existing basal activity, then pentameric channels with significant basal activity and biologically relevant functions should still be found in nature today, and indeed related GABA$_A$ receptors containing the β3 subunit display spontaneous/basal activity that is thought to control neuronal excitability[31].

In famous work involving the *lac* operon, Jacques Monod and François Jacob showed that gene expression can be controlled by a repressor protein that binds to DNA and inhibits constitutive expression of a target gene (Fig. 5a). Expression of the target gene could then be induced by inhibiting the repressor protein. In Monod's words, 'the inducer acts not by provoking' target gene expression, but instead 'by inhibiting an inhibitor' of its expression[32]. Similarly, we have shown that rather than 'provoking' AChR openings, agonists may instead 'inhibit the inhibition' of constitutive activity, and thus 'activation' of the AChR, the prototypical ligand-gated ion channel[33], may be the apparent manifestation of an analogous derepression mechanism (Fig. 5b).

## Methods

### Molecular biology

cDNA for the human AChR α1-subunit cloned into the pRBG4 plasmid was provided by Steven M. Sine, while cDNA encoding the ancestral β1 subunit (β$_{Anc}$) was ordered as a synthetic oligonucleotide and cloned into the same pRBG4 plasmid[15]. Point mutations substituting three arginine residues present in the homologous human 5-HT3$_A$ subunit[34] (Supplementary Fig. 1) into the human α1 subunit to produce the low conductance variant (α$_{LC}$; αD389R/N393R/A397R; mapped onto Protein Data Bank ID: 7QKO[35] in Supplementary Fig. 1) were introduced into cDNAs by inverse PCR[36]. Sanger sequencing confirmed the sequence of the entire reading frame.

### Mammalian cell expression

cDNAs encoding human and ancestral subunits were transfected into BOSC 23 cells[37] (CVCL_4401; ATCC number: CRL-11270). Cells were maintained in Dulbecco's modified Eagle's medium (DMEM) supplemented with 10% (by volume) fetal bovine serum at 37 °C, to a confluency of 50 to 70%. Calcium phosphate transfections were carried out for 3–4 h and terminated by media exchange. Experiments were performed 16–24 h after transfection. A separate plasmid encoding green fluorescent protein was included in all transfections to facilitate identification of transfected cells.

### Cell line authentication and mycoplasma testing

Approximately five million confluent cells were harvested and their total DNA isolated (E.Z.N.A.® Tissue DNA Kit), and then submitted to The Centre for Applied Genomics Genetic Analysis Facility (The Hospital for Sick Children, Toronto, Canada) for STR profiling using Promega's GenePrint® 24 System. A similarity search on the 8159 human cell lines with STR profiles in Cellosaurus release 42.0 was conducted on the resulting STR profile, which revealed that the cell line shares closest identity (88%, CLASTR 1.4.4 STR Similarity Search Tool score) with Anjou 65 (CVCL_3645). Anjou 65 is a child of CVCL_1926 (HEK293T/17) and is itself a parent line of CVCL_X852 (Bartlett 96). Bartlett 96 is the parent line of BOSC 23[37]. PCR tests confirmed that the cells were free from detectable mycoplasma contamination[38].

### Single-channel patch clamp recordings

Single-channel recordings from BOSC 23 cells transiently transfected with cDNAs encoding wild-type, ancestral, or low conductance subunits, were obtained in the cell-attached configuration with a membrane potential of −120 mV and a temperature maintained between 19 and 22 °C. The bath solution contained (in mM) 142 KCl, 5.4 NaCl, 0.2 CaCl$_2$ and 10 HEPES (4-(2-hydroxyethyl)−1-piperazineethanesulfonic acid), adjusted to pH 7.40 with KOH. The

pipette solution contained (in mM) 80 KF, 20 KCl, 40 K•aspartate, 2 MgCl$_2$, 1 EGTA (ethylene glycol-bis(β-aminoethyl ether)-*N,N,N′,N′*-tetraacetic acid), and 10 HEPES, adjusted to a pH of 7.40 with KOH. Acetylcholine chloride (Sigma) was added to pipette solutions to the desired final concentration and stored at −80 °C. Patch pipettes were fabricated from type 7052 or 8250 nonfilamented glass (King Precision Glass) with inner and outer diameters of 1.15 and 1.65 mm, respectively, and coated with SYLGARD 184 (Dow Corning). Prior to recording, electrodes were heat polished to yield a resistance of 5 to 8 MΩ. Single-channel currents were recorded using an Axopatch 200B patch clamp amplifier (Molecular Devices), with a gain of 100 mV/pA and an internal Bessel filter at 100 kHz. Data were sampled at 1.0 µs intervals using a BNC-2090 A/D converter with a National Instruments PCI 6111e acquisition card and recorded by the program Acquire (v. 6.0.0; Bruxton).

### Single-channel analysis

Single-channel event detection was performed using the program TAC (v. 4.3.3; Bruxton), where data were analysed with an applied 5 kHz digital Gaussian filter. Opening and closing transitions were detected using the 50% threshold crossing criterion, and corrected for instrument rise time using the R package *scbursts* (v. 1.6)[39]. Open and burst durations were placed into logarithmic bins[40], and the sum of exponentials was manually fit to the open and burst duration distributions within the program TACfit (v. 4.3.3; Bruxton). Bursts of closely spaced openings were defined by a critical closed duration (τ$_{crit}$), where individual openings separated by closings briefer than τ$_{crit}$ (2 ms throughout) were concatenated with their intervening closings to produce individual bursts[19]. Duration histograms were constructed using a universal 2 ms burst resolution, and by omitting bursts briefer than 100 µs. Supplementary Fig. 2 analysis includes burst open probability, which was determined within R (v. 4.0.4) using the packages *scburst* (v. 1.6)[39], *extremevalues* (v. 2.3.3)[41], and *MASS* (v. 7.3-55)[42].

### Electrical fingerprinting

Cells were transfected as described above, but with an α$_{LC}$ to β$_{Anc}$ ratio of 10:1, 1:1, or 0:1 (α$_{LC}$:β$_{Anc}$; by weight). Data were digitally filtered to 5 kHz within TAC and amplitudes were determined by setting the baseline and open-channel current for individual openings by eye. For each ratio, ~2000 events were collected and pooled from between 9–12 patches. A critical closed duration of 2 ms was uniformly applied throughout to define bursts of single-channel activity. To ensure that only fully resolved amplitudes were included in the analysis, events briefer than 100 µs were omitted. Pooled events were binned and plotted in event-based amplitude histograms, which were then fit with a set of Gaussian distributions. Data within 1.5 standard deviations of the mean amplitude encompassing each Gaussian, and representing each amplitude class, were then isolated for downstream dwell time analysis. Open and burst duration histograms were generated from events in each amplitude class in TACFit, and burst sorting and analysis was performed within R using the packages *scbursts* (v. 1.6)[39] and *MASS* (v. 7.3-55)[42].

### Paired recordings from the same cell

Paired recordings, in the absence (−) and then presence (+) of agonist, were obtained from successive cell-attached patches of the same cell. A 5 min continuous recording in the absence of agonist was obtained, and then a second 5 min recording from a new cell-attached patch, on the same cell, was acquired, but this time in the presence of 300 µM acetylcholine within the patch pipette. Data were filtered to 5 kHz, and all events within the 5 min window post voltage application were analysed. A critical closed duration and a burst resolution of 2 ms, was uniformly applied throughout. Paired data plots were generated in R with the packages *ggplot2* (v. 3.3.5)[43] and *ggpubr* (v. 0.4.0).

### Radioligand-binding experiments

AChR cell-surface expression was measured by binding of [$^{125}$I]-labelled α-bungarotoxin to transfected cells. Approximately 850,000 BOSC 23 cells were plated onto 6 cm dishes and transfected with α, β$_{Anc}$, or both α and β$_{Anc}$ (1:1; by weight) cDNA's. Cells were harvested one day post-transfection and incubated for 1 h at room temperature with [$^{125}$I]-α-bungarotoxin (25 nM; specific activity of 10 Ci/mmol) in potassium Ringer's solution. Ringer's solution contained (in mM), 140 KCl, 5.4 NaCl, 1.8 CaCl$_2$, 1.7 MgCl$_2$, 25 HEPES, and 30 mg L$^{-1}$ bovine serum albumin, adjusted to pH 7.40 with KOH[25]. Cells were deposited onto 25 mm Whatman GF/C microfiber filter discs using a Hoefer filtration manifold and washed three times with 5 mL of Ringer's solution. Nonspecific binding of [$^{125}$I]-α-bungarotoxin to the filter discs was minimised by pre-incubating each disc in Ringer's solution containing 1% bovine serum albumin for 1 h. Bound toxin was counted for 2 min in a Wizard2 1-Detector Gamma Counter (Perkin-Elmer).

### Statistics and reproducibility

For single-channel recordings, each replicate represents data acquired from the same cell-attached patch, where each patch was from a different cell. For the paired replicates in Fig. 4 and Supplementary Fig. 5, consecutive cell-attached patches, first in the absence and then in the presence of 300 μM acetylcholine, were acquired from the same cell. For radioligand-binding experiments, each replicate constitutes data acquired from separate transfections. Statistical tests were performed using GraphPad Prism (v. 8.0.0). One-way ANOVA (Fig. 2) was performed using Tukey's multiple comparison test. For paired replicates (Fig. 4 and Supplementary Fig. 5), separate two-way ANOVA tests were performed for each cDNA ratio. All significant differences had adjusted $p$-values below the predetermined α level of 0.05.

### Reporting summary

Further information on research design is available in the Nature Portfolio Reporting Summary linked to this article.

## Data availability

The data that support this study are available from the corresponding authors upon reasonable request. All source data associated with the current study are available in a figshare repository [https://doi.org/10.6084/m9.figshare.21742247]. This includes detected single-channel openings and closings ('evt' files also in 'txt' format), as well as raw data and any corresponding analysis scripts. Raw single-channel recordings in Bruxton Acquire format (*.acquire format; approximately 1 GB each) are available upon request. Supplementary Fig. 1 was made using 7QKO.

## Code availability

R scripts for defining and sorting bursts are freely available on figshare [https://doi.org/10.6084/m9.figshare.21742247].

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

## Acknowledgements

We thank Kathleen M. Gilmour for access to, and technical assistance with, a γ-counter. We thank Steven M. Sine, John E. Baenziger, and members of the dacosta]:[lab for comments on the manuscript. C.J.G.T. was funded in part by an Ontario Graduate Scholarship, while J.R.E. is the recipient of a Canada Graduate Scholarship from the Canadian Institutes of Health Research (CIHR) and a Natural Sciences and Engineering Research Council (NSERC) of Canada CREATE Scholarship. C.J.B.d.C. acknowledges grants from the Natural Sciences and Engineering Research Council of Canada (RGPIN-2016-04801), the Canada Foundation for Innovation (34475), the Canadian Institutes of Health Research (377068), as well as a New Frontiers in Research Fund-Exploration Grant (NFRFE-2018-00064).

## Author contributions

C.J.G.T. acquired and analysed all electrophysiological data, while R.M.S. and J.R.E. acquired preliminary α/β$_{Anc}$ heteromer recordings. J.R.E., C.J.B.d.C., and C.J.G.T. performed radiolabelled α-Btx experiments. C.J.G.T. and C.J.B.d.C. interpreted the data and wrote the manuscript. C.J.B.d.C. supervised the project.

## Competing interests

The authors declare no competing interests.
