## [Peer Review File · Nature Communications]

Derepression may masquerade as activation in ligand-gated ion channelsReviewers' Comments:

Reviewer #1:

Remarks to the Author:

Below are my comments; I have also attached a word document with identical content, formatted to be more readable.

A) Key results: Tessier et al thoroughly explore the properties of heteromeric muscle-type AChRs containing one or two modern human muscle-type alpha subunits, using radioligand binding, single-cell patch clamp, and electrical fingerprinting. Where the Banc homopentamer shows extended spontaneous bursts of channel activity, the introduction of successive alpha subunits into the heteromer reduces both the frequency and duration of these bursts, as well as the number of openings per burst, though the amplitudes are not affected. Addition of acetylcholine, which does not significantly alter the activity of the Banc homopentamer, does increase the frequency of activity bursts in the heteropentamer. This gives insight into the mechanism of channel function/activation (including of modern human subunits), and provides an intriguing experimental system with which to further probe these crucial questions about channel mechanics and activation.

B) Validity: The experimental design is thorough and consistent, the inferences drawn are robustly supported by the data, and the language used is generally precise. I have one major issue with the language used: the title of this work emphasises a difference between de-repression and activation. This either needs to be explored in more detail in the text, or the title (and phrasings throughout the text) should be changed to reflect more closely the contents of the work:

It is not clear from the text what the difference between derepression and activation means in an ion channel context. In the gene expression parallel, the gene has a basal activity level, but binding of an external factor can repress this, and removal of that external factor restores the basal activity. However, another gene expression paradigm is one where the basal expression level of the gene is low, and binding of an external factor (e.g. a transcription factor) increases the expression. In the gene example, then, the difference between derepression and activation is twofold: 1) derepression involves the removal of an external factor from the DNA, whereas activation involves the addition of one, and 2) the derepressible gene has a significant 'basal level' of activity (which can be repressed), whereas the activatable gene has low basal activity (which can be increased).

It is not clear to me that this distinction is reflected in the ion channel example, or indeed whether it can be unambiguously assigned.

Sub-point: The first difference named above, of external factors binding and unbinding, cannot be cleanly transferred to switching out of subunits for one another. An agonist is more clearly an external factor, but whether its binding to the channel counts as activation (binding of an external factor which causes measurable activity) or derepression (removing the inhibition of a repressor), is not obvious from the discussion here.

Sub-point: As for the second factor, if one counts the 'basal activity level' of the channel as being that of the Banc homopentamer, then it is significant, and the influence of alpha subunits in the heteropentamer would be repression (and agonist binding would cause derepression). However, if one counts the basal activity level as being that of the heteropentamer, then it is low, and the influence of more beta subunits in the homopentamer would be activation.

One might say that the Banc homopentamer should be assigned the 'basal' label, as it is hypothesised to be more ancient than the alpha subunits, in that case this should be stated in the text.

Additionally, to bring this concept beyond the theoretical: why would a difference between derepression and activation be important? What predictions can be made and tested on this basis (the testing itself would be outside the scope of this work). For example, with the alpha/Banc heteropentamer: if acetylcholine were activating it, then the channel should open in response to acetylcholine binding, whereas if it were simply derepressing, then one might expect more variable and longer times between ligand binding and channel opening. What other differences could be

predicted? Should there be any effect on amount of bursts vs open times vs openings per burst, or will they all have the same behaviour pattern? This is touched on in lines 195-213, but I find the phrasing used here somewhat imprecise or narrow:

Sub-point: 'Instead of activating openings, agonists could [...] allow [the channel] to open and close as dictated by its own intrinsic energy landscape'

Sub-sub-point: A closed channel that binds agonist to open is also dictated by its own intrinsic energy landscape, ie its own intrinsic energy landscape dictates it to be closed. Is the difference that in activation, agonist binding gives sufficient energy for the channel to overcome opening, whereas in derepression, agonist binding is necessary but not sufficient for channel opening, ie it only brings the channel to an energy level lower than the transition to open, and the energy to go the rest of the way must be acquired stochastically?

Sub-point: 'once derepressed, the channel opens and closes in the same way irrespective of the efficacy of the bound agonist'

Sub-sub-point: In comparison then, if agonists and partial agonists were activating (rather than derepressing) the channel, they would differ in their ability to cause channel opening. What are the broader consequences of this? Would one expect it to be observable on a population level, or are the implications here more applicable to understanding the mechanism itself, rather than macro-scale behaviours?

Other minor language/phrasing comments:

- Line 39: unclear from this sentence whether 'earlier stage in the activation process' means earlier than channel opening (from the broader discussion), or earlier than flipping/priming (which is the subject of the preceding sentence). Clarify what the origins of agonism are earlier than.

- Line 47-48: 'despite presumably being devoid of agonist-binding sites' is a strong statement. While the Banc subunits are shown not to bind alfa-bungarotoxin, and likely do not bind acetylcholine, that does not mean that the entire agonist binding site is lacking in the Banc homopentamer. More extensive assays of orthosteric ligand binding of a range of compounds would be needed to support this statement, and the statement should be moderated here.

- Line 110-111 'This control demonstrated that the conductance altering mutations did not affect other properties of the alfa subunit' goes further than the data shown. [...] 'did not measurably/significantly' (or some other alternative) would be more precise.

- Figure text figure 2: 'combination of the alfa subunit and Banc': consider substituting 'mixture' for 'combination', as the latter could imply chimeric subunits rather than the presence of both distinct subunits

- Figure text figure 2: the figure text mentions dashed lines, I cannot see any dashed lines in the figure.

- Figure text figure 3: 'shaded regions from inset 2 in (B))'. This presumably refers to shaded regions from inset 2 in (C).

C) Significance: This work gives insight into the mechanism of channel function, in particular channel activation, both from the perspective of evolutionary history (probing the roots of pLGIC formation and function) and towards understanding modern human pLGICs, which are crucial components of mammalian and non-mammalian nervous systems. Beyond the insight attained in this work itself, the system presented is shown to be a robust and pliable experimental system for probing pLGIC function, opening the way for a variety of further experiments.

D) Data and methodology: The data is robust and well presented, and the conclusions drawn are well supported by the experimental data, with the exception of the language around 'derepression versus activation', which I have requested clarification of above.

E) Analytical approach: The interpretation of the experimental data is well supported, and the statistical tests used are appropriate.

F) Suggested improvements: I have no suggested improvements to the experiments themselves. The

figures clearly and elegantly present the data, with one potential exception: the inset plots of amplitude vs log duration (fig 3C, fig S2) have amplitude on the y-axis, when the plots that they are insets of have amplitude along the x-axis. It might be easier to interpret the data and compare the inset plots to the surrounding data if they were oriented in the same way with respect to the amplitude. However, this is not a requirement for recommending publication of this work.

G) Clarity and context: The issue described in section about the use of 'derepression' versus 'activation' is currently a little unspecific. Apart from this, the text is clear, and leads the reader well through the concepts explored in the work.

H) References: The references are appropriate.

I) My expertise: My expertise lies in functional assays of pentameric ligand-gated ion channels, including single-cell patch clamp electrophysiology, and in the discussion of how their structure underpins their function. I have experience using and applying statistical tests to biological data.

Reviewer #2:

Remarks to the Author:

In this manuscript, Tessier and colleagues continue their distinctive work characterizing an ancestral homolog of nicotinic acetylcholine receptors, as a vehicle for larger questions about spontaneous and gated activity. Detailed but clear electrophysiology recordings and a cogent narrative elevate this story from the potentially convoluted or arcane, instead offering compelling insights into the mechanistic origins of, and parallels to, ligand activation. Though the language is carefully crafted and largely comprehensible, a few areas seem open to meaningful improvement.

First, it is interesting that -- despite the authors' careful consideration of complex physiological processes underlying their data -- they restrict themselves entirely to cartoon depictions of the interactions at play. Whereas sequence alignments of the constructs in this work have been previously reported, it would seem useful to update these with more detailed structural insights. Indeed, the recent publication of cryo-EM structures of closely related receptors -- including with bungarotoxin (e.g. Rahman et al., 2020) -- seem to offer a valuable opportunity to visualize the rationale for these experiments. Although detailed analysis of the structural or dynamic differences at alpha-beta, beta-alpha, and beta-beta interfaces may be the task of future study, it would seem worthwhile to include at least some representation of the relevant interaction surfaces presumed to account for e.g. the differential toxin and agonist binding in Figure 2, conductance-altering mutations applied in Figure 3, etc.

Second, given the considerable extent of the Results section -- including substantial interpretation and contextualization of the presented data -- it seems unusual for it to lack subheadings. Some content could be moved from Results to Discussion, although logical progression of the experiments does benefit from some of the detail currently included; a more standard approach would be to separate the Results into sections corresponding to the key figures (2, 3, 4).

As a minor additional comment, Figure 1 could be annotated in more detail. Presumably, upward deflection from baseline corresponds to inward cation currents, as is often (though not universally) the case in cell-attached recordings. Do the peripheries of burst activity, particularly in the apo β Anc homopentamer, correspond to initiation and termination of the -120 mV clamp?

Response to Review

Reviewers

Authors

Text changes

Reviewer #1 (Remarks to the Author):

A) Key results: Tessier et al. thoroughly explore the properties of heteromeric muscle-type AChRs containing one or two modern human muscle-type alpha subunits, using radioligand binding, single-cell patch clamp, and electrical fingerprinting. Where the Banc homopentamer shows extended spontaneous bursts of channel activity, the introduction of successive alpha subunits into the heteromer reduces both the frequency and duration of these bursts, as well as the number of openings per burst, though the amplitudes are not affected. Addition of acetylcholine, which does not significantly alter the activity of the Banc homopentamer, does increase the frequency of activity bursts in the heteropentamer. This gives insight into the mechanism of channel function/activation (including of modern human subunits), and provides an intriguing experimental system with which to further probe these crucial questions about channel mechanics and activation.

B) Validity: The experimental design is thorough and consistent, the inferences drawn are robustly supported by the data, and the language used is generally precise. I have one major issue with the language used: the title of this work emphasises a difference between de-repression and activation. This either needs to be explored in more detail in the text, or the title (and phrasings throughout the text) should be changed to reflect more closely the contents of the work:

It is not clear from the text what the difference between derepression and activation means in an ion channel context. In the gene expression parallel, the gene has a basal activity level, but binding of an external factor can repress this, and removal of that external factor restores the basal activity. However, another gene expression paradigm is one where the basal expression level of the gene is low, and binding of an external factor (e.g. a transcription factor) increases the expression. In the gene example, then, the difference between derepression and activation is twofold: 1) derepression involves the removal of an external factor from the DNA, whereas activation involves the addition of one, and 2) the derepressable gene has a significant 'basal level' of activity (which can be repressed), whereas the activatable gene has low basal activity (which can be increased).

It is not clear to me that this distinction is reflected in the ion channel example, or indeed whether it can be unambiguously assigned.

Sub-point: The first difference named above, of external factors binding and unbinding, cannot be cleanly transferred to switching out of subunits for one another. An agonist is more clearly an

external factor, but whether its binding to the channel counts as activation (binding of an external factor which causes measurable activity) or derepression (removing the inhibition of a repressor), is not obvious from the discussion here.

Sub-point: As for the second factor, if one counts the 'basal activity level' of the channel as being that of the Banc homopentamer, then it is significant, and the influence of alfa subunits in the heteropentamer would be repression (and agonist binding would cause derepression). However, if one counts the basal activity level as being that of the heteropentamer, then it is low, and the influence of more beta subunits in the homopentamer would be activation. One might say that the Banc homopentamer should be assigned the 'basal' label, as it is hypothesised to be more ancient than the alfa subunits, in that case this should be stated in the text.

Additionally, to bring this concept beyond the theoretical: why would a difference between derepression and activation be important? What predictions can be made and tested on this basis (the testing itself would be outside the scope of this work). For example, with the alpha/Banc heteropentamer: if acetylcholine were activating it, then the channel should open in response to acetylcholine binding, whereas if it were simply derepressing, then one might expect more variable and longer times between ligand binding and channel opening. What other differences could be predicted? Should there be any effect on amount of bursts vs open times vs openings per burst, or will they all have the same behaviour pattern? This is touched on in lines 195-213, but I find the phrasing used here somewhat imprecise or narrow: Sub-point: 'Instead of activating openings, agonists could [...] allow [the channel] to open and close as dictated by its own intrinsic energy landscape'

Sub-sub-point: A closed channel that binds agonist to open is also dictated by its own intrinsic energy landscape, ie its own intrinsic energy landscape dictates it to be closed. Is the difference that in activation, agonist binding gives sufficient energy for the channel to overcome opening, whereas in derepression, agonist binding is necessary but not sufficient for channel opening, ie it only brings the channel to an energy level lower than the transition to open, and the energy to go the rest of the way must be acquired stochastically?

Sub-point: 'once derepressed, the channel opens and closes in the same way irrespective of the efficacy of the bound agonist'

Sub-sub-point: In comparison then, if agonists and partial agonists were activating (rather than derepressing) the channel, they would differ in their ability to cause channel opening.

What are the broader consequences of this? Would one expect it to be observable on a population level, or are the implications here more applicable to understanding the mechanism itself, rather than macro-scale behaviours?

Sub-point: The first difference named above, of external factors binding and unbinding, cannot be cleanly transferred to switching out of subunits for one another. An agonist is more clearly an external factor, but whether its binding to the channel counts as activation (binding of an external

factor which causes measurable activity) or derepression (removing the inhibition of a repressor), is not obvious from the discussion here.

Sub-point: As for the second factor, if one counts the 'basal activity level' of the channel as being that of the Banc homopentamer, then it is significant, and the influence of alfa subunits in the heteropentamer would be repression (and agonist binding would cause derepression). However, if one counts the basal activity level as being that of the heteropentamer, then it is low, and the influence of more beta subunits in the homopentamer would be activation. One might say that the Banc homopentamer should be assigned the 'basal' label, as it is hypothesised to be more ancient than the alfa subunits, in that case this should be stated in the text.

Other minor language/phrasing comments:

- Line 39: unclear from this sentence whether 'earlier stage in the activation process' means earlier than channel opening (from the broader discussion), or earlier than flipping/priming (which is the subject of the preceding sentence). Clarify what the origins of agonism are earlier than.

- Line 47-48: 'despite presumably being devoid of agonist-binding sites' is a strong statement. While the Banc subunits are shown not to bind alfa-bungarotoxin, and likely do not bind acetylcholine, that does not mean that the entire agonist binding site is lacking in the Banc homopentamer. More extensive assays of orthosteric ligand binding of a range of compounds would be needed to support this statement, and the statement should be moderated here.

- Line 110-111 'This control demonstrated that the conductance altering mutations did not affect other properties of the alfa subunit' goes further than the data shown. [...] 'did not measurably/significantly' (or some other alternative) would be more precise.

- Figure text figure 2: 'combination of the alfa subunit and Banc': consider substituting 'mixture' for 'combination', as the latter could imply chimeric subunits rather than the presence of both distinct subunits

- Figure text figure 2: the figure text mentions dashed lines, I cannot see any dashed lines in the figure.

- Figure text figure 3: 'shaded regions from inset 2 in (B))'. This presumably refers to shaded regions from inset 2 in (C).

C) Significance: This work gives insight into the mechanism of channel function, in particular channel activation, both from the perspective of evolutionary history (probing the roots of pLGIC formation and function) and towards understanding modern human pLGICs, which are crucial components of mammalian and non-mammalian nervous systems. Beyond the insight attained in this work itself, the system presented is shown to be a robust and pliable experimental system for probing pLGIC function, opening the way for a variety of further experiments.

D) Data and methodology: The data is robust and well presented, and the conclusions drawn are well supported by the experimental data, with the exception of the language around 'derepression versus activation', which I have requested clarification of above.

E) Analytical approach: The interpretation of the experimental data is well supported, and the statistical tests used are appropriate.

F) Suggested improvements: I have no suggested improvements to the experiments themselves. The figures clearly and elegantly present the data, with one potential exception: the inset plots of amplitude vs log duration (fig 3C, fig S2) have amplitude on the y-axis, when the plots that they are insets of have amplitude along the x-axis. It might be easier to interpret the data and compare the inset plots to the surrounding data if they were oriented in the same way with respect to the amplitude. However, this is not a requirement for recommending publication of this work.

For clarity, below we repeat and address each point, one-by-one and in sequence, raised by Reviewer #1.

Validity: The experimental design is thorough and consistent, the inferences drawn are robustly supported by the data, and the language used is generally precise. I have one major issue with the language used: the title of this work emphasises a difference between de-repression and activation. This either needs to be explored in more detail in the text, or the title (and phrasings throughout the text) should be changed to reflect more closely the contents of the work:

We agree. As suggested by the reviewer we have revised (1) our title (see line 4), and (2) as outlined in the individual responses that follow, phrasings throughout the text.

It is not clear from the text what the difference between derepression and activation means in an ion channel context.

We agree that the difference between activation and derepression in the context of an ion channel needs to be made explicitly clear. Our revised manuscript, and the detailed response presented here, clarify this important distinction.

Activation in an ion channel context

In its most general form, which derives from the classic pharmacological perspective, 'activation' refers to the transition of a receptor from an *inactive* to an *active* state following application of

agonist. Implied within this broad definition of ‘activation’ is the idea that the agonist molecules directly drive the transition of the receptor from an *inactive* to an *active* state.

For ion channels, their open/ion-conducting conformation is generally thought of as their *active* state, and ion channel ‘activation’ is considered the transition from a closed to an open state, and thus opening of the ion channel driven by an external stimulus. For ligand-gated (or ‘agonist-activated’) ion channels, agonists are expected to directly drive this transition from an *inactive/closed* state to an *active/open* state. In essence, the free energy liberated upon the binding of agonist to the channel is used to stabilize the channel’s *active/open* state. Accordingly, agonists with different chemical structures, and thus different physical interactions with the ion channel, should differ in their ability to stabilize the *active/open* state.

Derepression in an ion channel context

From the prototypical gene expression perspective, derepression is the removal of a repressive influence, leading to an ‘apparent activation’, which is in fact the restoration of intrinsic basal activity. In an ion channel context, to derepress an ion channel, once bound ‘agonists’ (note the quotations) should release, or drive the transition of, the channel from a *repressed/closed* state, where openings are unlikely, to a *derepressed/closed* state where openings are more likely. An important distinction between derepression and activation of an ion channel, is that in the derepression paradigm, ‘agonists’ do not drive the transition from an *inactive/closed* state to an *active/open* state, but instead drive a transition between two distinct closed states that are both *inactive/closed*. This derepression mechanism predicts that the ultimate opening of the ion channel should be independent of the ‘agonist’ because the free energy of binding is used to stabilize the channel’s *derepressed/closed* state, as opposed to its *active/open* state. This lack of dependence of ion channel opening on the agonist is precisely what has been realized with the modern ‘flip’ and ‘prime’ models of ‘activation’.

Our revised manuscript contains an abbreviated version of the above in the Discussion section (see lines 267–300 of revised manuscript).

In the gene expression parallel, the gene has a basal activity level, but binding of an external factor can repress this, and removal of that external factor restores the basal activity. However, another gene expression paradigm is one where the basal expression level of the gene is low, and binding of an external factor (e.g. a transcription factor) increases the expression. In the gene example, then, the difference between derepression and activation is twofold: 1) derepression involves the removal of an external factor from the DNA, whereas activation involves the addition of one, and 2) the derepressable gene has a significant ‘basal level’ of activity (which can be repressed), whereas the activatable gene has low basal activity (which can be increased).

It is not clear to me that this distinction is reflected in the ion channel example, or indeed whether it can be unambiguously assigned.

We agree that assigning whether a channel has significant basal activity (spontaneous opening in the absence of agonist) that can be repressed (and then derepressed upon “activation”) versus a channel having low basal activity that can be increased (upon *bona fide* activation) is non-trivial, and we do not claim to have unambiguously discriminated between these two possibilities for the AChR.

We propose that the present work exposes the possibility that the α -subunits directly repress basal activity of the AChR. This claim stems from insights gleaned from our system involving β_{Anc} , the reconstructed ancestral β -subunit, which forms homopentamers with significant basal activity (spontaneous opening). The resemblance of β_{Anc} basal activity to that of the “agonist-activated” AChR is uncanny, suggesting that rather than being fortuitous, it reflects a fundamental property of AChR subunits that is deeply embedded in their sequence, and thus structure, function, and evolutionary history. This β_{Anc} subunit also exhibits remarkable plasticity, as it not only forms homopentamers, but can also mix with extant subunits to form heteropentamers with a variety of subunit stoichiometries and arrangements. In the present work, this allowed us to probe the function of the human α -subunit in a unique context. We demonstrated that swapping in a human α -subunit for a single β_{Anc} subunit led to a dramatic decrease of basal activity. This decrease was even more pronounced when a second α -subunit was swapped in. The important point is that the basal activity of the β_{Anc} -homopentamers is high, and that of the α/β_{Anc} heteromers is low, with the only difference between them being the absence or presence of one or two α -subunits (at the expense of one or two β_{Anc} subunits). Swapping in α -subunits decreases basal activity, and as mentioned in our original submission:

“Whether this is through actively repressing spontaneous openings, or by failing to contribute energetically to them in the same way that the β_{Anc} subunit they replace would, the result is the same: spontaneous openings are repressed in α/β_{Anc} heteromers.”

As per the reviewer’s comment, our revised manuscript no longer contains the above sentence, and instead contains the more explicit discussion below (see lines 253–259 of revised manuscript):

“Relative to β_{Anc} homopentamers, the reduced spontaneous activity of α/β_{Anc} heteromers depends steeply upon the number of incorporated α -subunits, and thus it appears as though the α -subunits have the capacity to repress intrinsic spontaneous activity. However, a similar outcome would be expected if each β_{Anc} subunit imparted an activating presence that is lost in α/β_{Anc} heteromers. Determining the relative contribution of these two possibilities is not trivial, yet the implications of a possible repressive role for the α -subunits are profound.”

Sub-point: The first difference named above, of external factors binding and unbinding, cannot be cleanly transferred to switching out of subunits for one another.

We agree that there is a difference between external factors binding and unbinding versus switching out of subunits for one another. The complication is the one discussed above. Specifically, that the effect of switching subunits could be the result of function *gained* by the presence of the 'switched in' subunit, or of function *lost* due to the absence of the 'switched out' subunit.

We also agree that, with exception of the γ/ϵ (fetal/adult) subunit swap, it is not possible to cleanly switch out subunits in the modern-day muscle-type AChR for the purpose of measuring their influence on basal activity. This limitation is precisely what our β_{Anc} -based system allowed us to overcome. In the present work we 'switched out' β_{Anc} subunits for α -subunits, and thus directly measured the influence of each successive α -subunit on basal activity. Granted, this is not simply the addition of one or two α -subunit(s), but also the corresponding subtraction of one or two β_{Anc} subunit(s). Nevertheless, as clarified in our revised statement above, the result is the same: basal activity is decreased in α/β_{Anc} heteromers relative to β_{Anc} homopentamers.

An agonist is more clearly an external factor, but whether its binding to the channel counts as activation (binding of an external factor which causes measurable activity) or derepression (removing the inhibition of a repressor), is not obvious from the discussion here.

We agree that discriminating between these two possibilities is challenging. To our knowledge, a derepression mechanism has not been conceived of, or formally proposed, let alone demonstrated, for any class of ligand-gated ion channel. To demonstrate that a derepression mechanism is plausible, it is necessary to first demonstrate the possible presence of a repressor. A novel aspect of the present work is that it exposes a possible repressive role for the α -subunits, and therefore that 'agonists' could work through such a derepression mechanism.

Sub-point: As for the second factor, if one counts the 'basal activity level' of the channel as being that of the β_{Anc} homopentamer, then it is significant, and the influence of α subunits in the heteropentamer would be repression (and agonist binding would cause derepression). However, if one counts the basal activity level as being that of the heteropentamer, then it is low, and the influence of more β subunits in the homopentamer would be activation.

We agree that the point of reference is important for clarity (less so for validity – see below), and for this reason, we were explicit in our original submission:

“From the perspective of β_{Anc} homopentamers, the above data demonstrate that incorporation of human α -subunits leads to a repression of spontaneous single-channel activity.”

We also agree that the scenarios presented by the reviewer, where either α -subunits ‘repress’ or β_{Anc} subunits ‘activate’, are both possible, and that this original statement presumed that incorporation of the α -subunits led to repression. Furthermore, although we believe it was implied, we recognize that this statement failed to mention that β_{Anc} subunits were also ‘switched out’ in these experiments. To reflect the two possible interpretations raised by the reviewer, and further clarify this statement, we have made it more precise in lines 202–204 of our revised manuscript:

“From the perspective of β_{Anc} homopentamers, the above data demonstrate that **replacing one or more β_{Anc} -subunit(s) with the human α -subunit** leads to a **reduction in** spontaneous single-channel activity.”

One might say that the Banc homopentamer should be assigned the ‘basal’ label, as it is hypothesised to be more ancient than the alfa subunits, in that case this should be stated in the text.

As discussed in one of our previous publications, there is no obvious evolutionary argument as to why β_{Anc} should form homopentamers, and further that these homopentamers would spontaneously open (PMID: 35781368). At the same time, we believe whether β_{Anc} homopentamers or α/β_{Anc} heteromers are assigned the ‘basal’ level does not matter. As the reviewer points out, from the perspective of α/β_{Anc} heteromers the influence of more β_{Anc} subunits could be activating. The inverse is also possible. Subtraction of a repressive α -subunit could manifest as apparent activation. The important point is that it is not trivial to tell which mechanism is at play, but both are possible.

The present work, beginning with β_{Anc} homopentamers and moving to α/β_{Anc} heteromers, exposed a possible repressive role for the α -subunits. As the reviewer points out, this interpretation could depend upon the frame of reference. To present a more balanced perspective, in our revised manuscript (1) we use the term ‘apparent repression’ when describing our results, (2) we explicitly mention the alternate interpretation, and (3) we restrict our development of a derepression mechanism to the Discussion section.

Additionally, to bring this concept beyond the theoretical: why would a difference between derepression and activation be important? What predictions can be made and tested on this basis (the testing itself would be outside the scope of this work). For example, with the alpha/Banc heteropentamer: if acetylcholine were activating it, then the channel should open in response to acetylcholine binding, whereas if it were simply derepressing, then one might expect

more variable and longer times between ligand binding and channel opening. What other differences could be predicted? Should there be any effect on amount of bursts vs open times vs openings per burst, or will they all have the same behaviour pattern? This is touched on in lines 195-213, but I find the phrasing used here somewhat imprecise or narrow: Sub-point: 'Instead of activating openings, agonists could [...] allow [the channel] to open and close as dictated by its own intrinsic energy landscape'

Sub-sub-point: A closed channel that binds agonist to open is also dictated by its own intrinsic energy landscape, ie its own intrinsic energy landscape dictates it to be closed.

We agree. To avoid confusion, in our revised manuscript we have removed the statement in question.

Is the difference that in activation, agonist binding gives sufficient energy for the channel to overcome opening, whereas in derepression, agonist binding is necessary but not sufficient for channel opening, ie it only brings the channel to an energy level lower than the transition to open, and the energy to go the rest of the way must be acquired stochastically?

Essentially, yes. However, much more work would have to be done to characterize the energy landscapes. In lines 283–289 of our revised manuscript we have tried to clarify this, and at the same time keep the discussion conceptual:

“To derepress the channel, agonists drive the transition of the channel from a repressed/closed state where openings are unlikely (e.g. ‘unflipped’ or ‘unprimed’), to a derepressed/closed state where openings are more likely (e.g. ‘flipped’ or ‘primed’). An important distinction between derepression and activation, is that in the derepression paradigm, agonists do not drive the transition from an inactive/closed state to an active/open state, but instead drive a transition between two distinct closed states that are both inactive.”

Sub-point: 'once derepressed, the channel opens and closes in the same way irrespective of the efficacy of the bound agonist'

Sub-sub-point: In comparison then, if agonists and partial agonists were activating (rather than derepressing) the channel, they would differ in their ability to cause channel opening.

Precisely. Based upon the definition of activation that we now present, agonists should directly affect the transition of the channel from an *inactive*/closed state to an *active*/open state. An extension being that agonists with different chemical structures, and thus different physical interactions with the ion channel, should also differ in their ability to stabilize the *active*/open state, which, based on the 'flip' and 'prime' models, we know is not the case. In our revised

Discussion we explicitly mention how a derepression paradigm reconciles these counterintuitive findings and provides insight into ‘flipped’ and ‘primed’ states that to this point have remained largely phenomenological. We believe this is one of the more important implications of the present work (starting on line 289 of our revised manuscript).

“From this perspective, full versus partial agonists differ in their ability to derepress the channel, but once derepressed, the channel opens and closes in the same way irrespective of the efficacy of the bound agonist. These predictions that (1)... and (2) channel opening is independent of agonist efficacy, are both realized in the ‘flip’ and ‘prime’ models.”

What are the broader consequences of this? Would one expect it to be observable on a population level, or are the implications here more applicable to understanding the mechanism itself, rather than macro-scale behaviours?

We believe there are several implications of a derepression mechanism. As the reviewer points out, these are ‘more applicable to understanding the mechanism (of activation) itself, rather than macro-scale behaviours.’ The main implications we now highlight in our revised manuscript are:

1. The ability of a derepression mechanism to reconcile the counterintuitive observation stemming from both the ‘flip’ and ‘prime’ models that different agonists do not lead to different opening rates of the channel. By reconciling this we believe that the derepression paradigm also provides insight in the ‘flipped’ and ‘primed’ states, which to this point have remained largely phenomenological.
 2. A derepression mechanism, where inhibition of an inhibitor by binding of an inducer (i.e., agonist) is layered on top of pre-existing intrinsic basal activity, also suggests a plausible path by which ‘agonist activation’ could have evolved in ligand-gated ion channels. While we mention this in the Discussion, this will need to be fleshed out in future work.
-

Other minor language/phrasing comments:

- Line 39: unclear from this sentence whether ‘earlier stage in the activation process’ means earlier than channel opening (from the broader discussion), or earlier than flipping/priming (which is the subject of the preceding sentence). Clarify what the origins of agonism are earlier than.

Thank you. We agree and have revised this sentence to (see lines 63–66):

“These findings placed the origins of agonism **before channel opening** in the **overall** activation process (Lape et al., 2008), and demonstrated that the ultimate opening and closing rates of the

AChR were independent of the agonist used to elicit them; a somewhat counterintuitive finding for a channel 'activated' by agonist."

- Line 47-48: 'despite presumably being devoid of agonist-binding sites' is a strong statement. While the Banc subunits are shown not to bind alfa-bungarotoxin, and likely do not bind acetylcholine, that does not mean that the entire agonist binding site is lacking in the Banc homopentamer. More extensive assays of orthosteric ligand binding of a range of compounds would be needed to support this statement, and the statement should be moderated here.

We agree. The validity of the present work does not depend on this assertion, and so to be conservative, in our revised manuscript we have removed this statement (see line 75 of revised manuscript).

- Line 110-111 'This control demonstrated that the conductance altering mutations did not affect other properties of the alfa subunit' goes further than the data shown. [...] 'did not measurably/significantly' (or some other alternative) would be more precise.

We agree. As suggested by the reviewer we have revised our text to:

"This control demonstrated that the conductance altering mutations did not **measurably** affect other properties of the α -subunit." (See line 163 of revised manuscript)

- Figure text figure 2: 'combination of the alfa subunit and Banc': consider substituting 'mixture' for 'combination', as the latter could imply chimeric subunits rather than the presence of both distinct subunits

We agree and have made the following change in the caption to Figure 2.

"...or a 1:1 (by weight) **mixture** of the α -subunit and β_{Anc} (1:1; intermediate grey)."

(See line 624 of revised manuscript)

- Figure text figure 2: the figure text mentions dashed lines, I cannot see any dashed lines in the figure.

We apologize for the confusion. The dashed lines in the top histogram of panel 'd' are hidden by the solid line of the overall fit. This is a result of the fact that there is only a single exponential component, which therefore overlaps perfectly with the overall fit. To avoid confusion, we explicitly added the sentence below to the end of the Figure 2 caption:

“Note that the dashed lines in the top panel are hidden by the solid line, as there is only a single exponential component that overlaps perfectly with the overall fit.”

(See lines 646 – 647 of revised manuscript)

- Figure text figure 3: 'shaded regions from inset 2 in (B)'. This presumably refers to shaded regions from inset 2 in (C).

The reviewer is correct. This refers to the shaded regions from inset 2 in panel “c”. We have corrected this in the Figure 3 caption in our revised manuscript. Thank you.

C) Significance: This work gives insight into the mechanism of channel function, in particular channel activation, both from the perspective of evolutionary history (probing the roots of pLGIC formation and function) and towards understanding modern human pLGICs, which are crucial components of mammalian and non-mammalian nervous systems. Beyond the insight attained in this work itself, the system presented is shown to be a robust and pliable experimental system for probing pLGIC function, opening the way for a variety of further experiments.

D) Data and methodology: The data is robust and well presented, and the conclusions drawn are well supported by the experimental data, with the exception of the language around 'derepression versus activation', which I have requested clarification of above.

E) Analytical approach: The interpretation of the experimental data is well supported, and the statistical tests used are appropriate.

F) Suggested improvements: I have no suggested improvements to the experiments themselves. The figures clearly and elegantly present the data, with one potential exception: the inset plots of amplitude vs log duration (fig 3C, fig S2) have amplitude on the y-axis, when the plots that they are insets of have amplitude along the x-axis. It might be easier to interpret the data and compare

the inset plots to the surrounding data if they were oriented in the same way with respect to the amplitude. However, this is not a requirement for recommending publication of this work.

We agree that this is less than ideal and appreciate the reviewer's suggestion. The problem stems from the fact that both quantities (duration and amplitude) in Figure 3, panel c, inset 1, are typically plotted on the x-axes in isolated duration or event-based amplitude histograms. However, for an amplitude vs. duration plot, which attempts to show a correlation between the two quantities, one of the quantities must be plotted on the y-axis. While we could plot the associated event-based amplitude histograms in panel c vertically (i.e., with amplitude on the y-axis, as in inset 1), we have found that this makes it difficult to clearly see the three amplitude components, as well as how they are broken down in the individual mixing experiments (i.e., inset 2). Essentially to do this, we would have to stretch this panel vertically, making the figure excessively tall. For these reasons, we prefer to leave this figure as is.

G) Clarity and context: The issue described in section about the use of 'derepression' versus 'activation' is currently a little unspecific. Apart from this, the text is clear, and leads the reader well through the concepts explored in the work.

H) References: The references are appropriate.

I) My expertise: My expertise lies in functional assays of pentameric ligand-gated ion channels, including single-cell patch clamp electrophysiology, and in the discussion of how their structure underpins their function. I have experience using and applying statistical tests to biological data.

Reviewer #2 (Remarks to the Author):

In this manuscript, Tessier and colleagues continue their distinctive work characterizing an ancestral homolog of nicotinic acetylcholine receptors, as a vehicle for larger questions about spontaneous and gated activity. Detailed but clear electrophysiology recordings and a cogent narrative elevate this story from the potentially convoluted or arcane, instead offering compelling insights into the mechanistic origins of, and parallels to, ligand activation. Though the language is carefully crafted and largely comprehensible, a few areas seem open to meaningful improvement.

First, it is interesting that -- despite the authors' careful consideration of complex physiological processes underlying their data -- they restrict themselves entirely to cartoon depictions of the interactions at play. Whereas sequence alignments of the constructs in this work have been previously reported, it would seem useful to update these with more detailed structural insights. Indeed, the recent publication of cryo-EM structures of closely related receptors -- including with bungarotoxin (e.g. Rahman et al., 2020) -- seem to offer a valuable opportunity to visualize the rationale for these experiments. Although detailed analysis of the structural or dynamic differences at alpha-beta, beta-alpha, and beta-beta interfaces may be the task of future study, it would seem worthwhile to include at least some representation of the relevant interaction surfaces presumed to account for e.g. the differential toxin and agonist binding in Figure 2, conductance-altering mutations applied in Figure 3, etc.

We agree. The supplemental information of our revised manuscript now contains a figure (new Figure S1) depicting a sequence alignment and the location of the residues that were substituted to produce the conductance altering mutant (α_{LC}) mapped onto a recent AChR structure.

Second, given the considerable extent of the Results section -- including substantial interpretation and contextualization of the presented data -- it seems unusual for it to lack subheadings. Some content could be moved from Results to Discussion, although logical progression of the experiments does benefit from some of the detail currently included; a more standard approach would be to separate the Results into sections corresponding to the key figures (2, 3, 4).

We agree. We now include subheadings in our revised Results section that correspond to the key figures. These subheadings are:

Line 92 (Fig. 2a): *The human α -subunit coassembles with β_{Anc}*

Line 118 (Fig. 2b-d): *Incorporation of α -subunits reduces spontaneous activity*

Line 146 (Fig. 3): *Reduction in spontaneous activity depends on the number of α -subunits*

Line 201 (Fig. 4): *Agonist relieves apparent repression of spontaneous activity*

As a minor additional comment, Figure 1 could be annotated in more detail. Presumably, upward deflection from baseline corresponds to inward cation currents, as is often (though not universally) the case in cell-attached recordings. Do the peripheries of burst activity, particularly in the apo β_{Anc} homopentamer, correspond to initiation and termination of the -120 mV clamp?

We agree and have revised the Figure 1 caption (as well as the other figure captions describing single-channel data). The caption now contains specific reference to the openings representing inward cation currents, and the -120 mV being constantly applied.

(See lines 602–694 of revised manuscript)

- END -

Reviewers' Comments:

Reviewer #1:

Remarks to the Author:

The work in question is thorough, clearly presented, and exciting. The added discussion on distinguishing activation and derepression in an ion channel context helps the reader understand the impact of this work.